# Structure-Preserving Embedding of Multi-layer Networks

## Abstract

This paper investigates structure-preserving embedding for multi-layer networks with community structure. We propose a novel generative tensor-based latent space model (TLSM) that allows heterogeneity among vertices. It embeds vertices into a low-dimensional latent space so that vertices within the same community are close to each other in the ambient space, and captures layer heterogeneity through a layer-effect factor matrix. With a general and flexible tensor decomposition on the expected network adjacency tensor, TLSM is dedicated to preserving the original vertex relations and layer-specific effects in the network embedding. An efficient alternative updating scheme is developed to estimate the model parameters and conduct community detection simultaneously. Theoretically, we establish the asymptotic consistencies of TLSM in terms of both multi-layer network estimation and community detection. The theoretical results are supported by extensive numerical experiments on both synthetic and real-life multi-layer networks.

## 1 Introduction

Network has arisen as one of the most common structures to represent the relations among entities. In many complex systems, entities can be multi-relational in that they may interact with each other under various circumstances. A multi-layer network, which consists of a common vertex set across all network layers representing the entities and an edge set at each layer to characterize a particular type of relation among entities, is faithful to represent these relations. Examples of multi-layer networks include social networks of multiple interaction channels [42, 15], biological networks of different collaboration schemes [49, 31, 29] and world trading networks [1, 37] of various goods.

In this paper, we propose a structure-preserving embedding framework for multi-layer networks via a tensor-based latent space model. Specifically, TLSM utilizes the factorization of network adjacency tensor as a building block, embeds the vertices into a low dimensional latent space, and captures the heterogeneity among different layers through a layer-effect factor matrix. Consequently, the community structure of the multi-layer network can be detected from a network embedding perspective, such that vertices within the same community are closer to one another in the ambient space than those in different communities. In addition, one key feature of TLSM is that it introduces a sparsity factor into the vanilla logit transformation of the network adjacency tensor, which allows TLSM to model sparse multi-layer networks in a more explicit fashion and accommodate relatively sparser multi-layer networks as the ones considered in literature [22]. More importantly, this sparsity factor can be estimated from the network adjacency tensor directly.

The main contribution of this paper is three-fold. First, the proposed TLSM is flexible and general in that it includes many popular network models as special cases. It also relaxes the layer-wise positive semi-definite condition that has been frequently employed in literature [6, 35]. Second, a joint modeling framework is constructed for TLSM, consisting of the multi-layer network likelihood

and a clustering type penalty, to estimate the multi-layer network and conduct community detection simultaneously. Its advantages are supported by extensive numerical experiments on both synthetic and real-life multi-layer networks. Third, the asymptotic consistencies of TLSM are established in terms of both multi-layer network estimation and community detection. Notably, the established theoretical results imply that the proposed methods can accommodate the sparsest multi-layer networks considered in literature.

The rest of the paper is organized as follows. The remaining of Section 1 discusses related works and introduces necessary notations. Section 2 presents the proposed TLSM and its estimation scheme with an efficient algorithm. In Section 3, we establish the asymptotic consistencies of TLSM. Extensive numerical performance of TLSM on synthetic and real-life multi-layer networks as well as ablation studies on two novel components of the proposed method are carried out in Section 4. Section 5 concludes the paper. The supplementary materials contains technique proofs and necessary lemmas, additional simulation studies, detailed parameter tuning process, among others.

## 1.1 Related work

While there is a growing number of literature focusing on community detection in single-layer network [48, 28, 13], community detection in multi-layer network is still in its infancy. One classical approach is to detect community structure in each layer separately [4, 5], which fails to leverage the homogeneity across different layers. Another approach is to aggregate multi-layer networks into a single-layer one [41, 12, 35], which heavily relies on the assumption of homogeneous linking pattern across multiple layers. Recently, [26] proposed to aggregate the biased-adjusted version of the squared adjacency matrix in each layer to alleviate the information loss in aggregation. yet it requires the average node degree to grow at a sub-optimal order.

In terms of multi-layer network generative models, [34] extended the seminal stochastic block model (SBM; 19) to the multi-layer stochastic block model (MLSBM; 34), where the probability for any two vertices to form an edge in a given layer depends only on their community memberships. Clearly, MLSBM heavily relies on the assumption of homogeneous vertices within communities. The framework of MLSBM has also been incorporated in degree-corrected network estimation [36], spectral clustering [6, 35, 26], least square estimation [27] and likelihood-based approaches [45]. In addition, network response regression model [46] and tensor factorization methods [8, 22] have also been proposed to detect community structures in multi-layer networks.

To allow heterogeneous vertices, the latent space model [18] and random dot product graph model [3] have been extended to multi-layer networks[47, 32, 2]. In addition, graph neural network and graph convolutional networks has been extended to multi-layer network for learning the multi-layer network embedding [14, 23, 17, 39].

## 1.2 Notations

Throughout the paper, we use boldface calligraphic Euler scripts ($\mathcal{A}$) to denote tensors, boldface capital letters ($\boldsymbol{A}$) or Greece letters ($\boldsymbol{\alpha}, \boldsymbol{\beta}$) to denote matrices, boldface lowercase letters ($\boldsymbol{a}$) to denote vectors, and regular letters ($a$) to denote scalars. For an order three tensor $\mathcal{A} \in \mathbb{R}^{I_1 \times I_2 \times I_3}$, $\mathcal{A}_{i,:,:} \in \mathbb{R}^{I_2 \times I_3}, \mathcal{A}_{:,j,:} \in \mathbb{R}^{I_1 \times I_3}$, and $\mathcal{A}_{:,:,m} \in \mathbb{R}^{I_1 \times I_2}$ are the $i$-th horizontal slide, $j$-th lateral slide and $m$-th frontal slide of $\mathcal{A}$, respectively. Similarly, for a matrix $\boldsymbol{A}$, $\boldsymbol{A}_{i,:}$ denotes its $i$-th row and $\boldsymbol{A}_{:,j}$ denotes its $j$-th column. For a vector $\boldsymbol{a}$, $\mathrm{diag}(\boldsymbol{a})$ stands for the diagonal matrix whose diagonal is $\boldsymbol{a}$. We use $||\cdot||, ||\cdot||_\infty$, and $||\cdot||_F$ to denote the $l_2$-norm, $l_\infty$-norm of a vector, and the Frobenius norm of matrix or tensor, respectively. For any integer $n$, denote $[n] = \{1, 2, ..., n\}$.

The mode-1 product between a tensor $\mathcal{A} \in \mathbb{R}^{I_1 \times I_2 \times I_3}$ and a matrix $\boldsymbol{U} \in \mathbb{R}^{J_1 \times I_1}$ is a tensor $\mathcal{A} \times_1 \boldsymbol{U} \in \mathbb{R}^{J_1 \times I_2 \times I_3}$ such that its $(j_1, i_2, i_3)$-th entry is defined as $(\mathcal{A} \times_1 \boldsymbol{U})_{j_1, i_2, i_3} = \sum_{i_1=1}^{I_1} \mathcal{A}_{i_1, i_2, i_3} \boldsymbol{U}_{j_1, i_1}$. The mode-2 or mode-3 product between $\mathcal{A}$ and any matrix of appropriate dimension are defined similarly. The CANDECOMP/PARAFAC (CP) decomposition of $\mathcal{A}$ has the form

$$\mathcal{A} = \sum_{r=1}^{R} \boldsymbol{a}^{(r)} \circ \boldsymbol{b}^{(r)} \circ \boldsymbol{c}^{(r)}, \tag{1}$$

where $\boldsymbol{a}^{(r)} \in \mathbb{R}^{I_1}, \boldsymbol{b}^{(r)} \in \mathbb{R}^{I_2}$, and $\boldsymbol{c}^{(r)} \in \mathbb{R}^{I_3}$ for $r \in [R]$, and $\circ$ stands for the vector outer product. The CP-rank [24] of the tensor $\boldsymbol{a}^{(r)} \circ \boldsymbol{b}^{(r)} \circ \boldsymbol{c}^{(r)}$ is defined to be 1, for $r \in [R]$. The minimal number

of rank-1 tensors in the CP decomposition of $\mathcal{A}$ is called the CP-rank of $\mathcal{A}$. Let $\mathcal{I} \in \{0,1\}^{R \times R \times R}$ be the identity tensor such that $\mathcal{I}_{i_1,i_2,i_3} = 1$ if $i_1 = i_2 = i_3$ and 0 otherwise, and let $A \in \mathbb{R}^{I_1 \times R}$, $B \in \mathbb{R}^{I_2 \times R}$, and $C \in \mathbb{R}^{I_3 \times R}$ such that $A_{\cdot,r} = a^{(r)}$, $B_{\cdot,r} = b^{(r)}$, and $C_{\cdot,r} = c^{(r)}$. Equation (1) then can be equivalently written as $\mathcal{A} = \mathcal{I} \times_1 A \times_2 B \times_3 C$.

# 2 Structure-preserving embedding

In this paper, we consider multi-layer networks that can be represented as an undirected and un-weighted $M$-layer graph $\mathcal{G} = (V, \mathcal{E})$, where $V = [n]$ consists of the common $n$ vertices across different layers, and $\mathcal{E} = \{E^{(m)}\}_{m=1}^M$ with $E^{(m)} \subset V \times V$ representing the $m$-th relation network among vertices. A order three adjacency tensor $\mathcal{A} = (a_{i,j,m}) \in \{0,1\}^{n \times n \times M}$ is then defined to represent $\mathcal{G}$ with entries $a_{i,j,m} = 1$ if $(i,j) \in E^{(m)}$ and 0 otherwise.

## 2.1 Tensor-based latent space model

To fully characterize the multi-layer network structure, we propose the following generative tensor-based latent space model (TLSM). For any $i \leq j \in [n]$, and $m \in [M]$,

$$a_{i,j,m} = a_{j,i,m} \overset{ind.}{\sim} \text{Bernoulli}(p_{i,j,m}), \text{ with} \tag{2}$$

$$\theta_{i,j,m} = \log\left(\frac{p_{i,j,m}}{s_n - p_{i,j,m}}\right), \text{ and} \tag{3}$$

$$\Theta = \mathcal{I} \times_1 \alpha \times_2 \alpha \times_3 \beta, \ \alpha \in \Omega_{\alpha}, \beta \in \Omega_{\beta}, \tag{4}$$

where $\mathcal{I}$ is the order three $R$-dimensional identity tensor. Basically, (2) follows the standard routine in the multi-layer network literature [34, 35, 27, 22] to model that $a_{i,j,m} = a_{j,i,m}$ are independently generated from a Bernoulli distribution, for $i \leq j \in [n]$ and $m \in [M]$. Denote $\mathcal{P} = (p_{i,j,m}) \in \mathbb{R}^{n \times n \times M}$ as the network underlying probability tensor, and then $\Theta = (\theta_{i,j,m}) \in \mathbb{R}^{n \times n \times M}$ is the entry-wise transformation of $\mathcal{P}$ by (3). We call the transformation (3) as the modified logit transformation in that the constant 1 in the standard logit transformation is replaced by a sparsity factor $s_n$, which may vanish with $n$ and $M$. We further assume all entries of $\mathcal{P}$ are of the order $s_n$; that is, there exists a constant $\frac{1}{2} \leq \xi < 1$ such that $(1-\xi)s_n \leq p_{i,j,m} \leq \xi s_n$, for $i,j \in [n]$ and $m \in [M]$. Thus, $s_n$ essentially controls the overall network sparsity and the entries of $\Theta$ are ensured to locate in the interval $[-\log\frac{\xi}{1-\xi}, \log\frac{\xi}{1-\xi}]$. More importantly, (4) models the CP decomposition of $\Theta$ by the factor matrices $\alpha \in \mathbb{R}^{n \times R}$ and $\beta \in \mathbb{R}^{M \times R}$ with CP-rank $R$, which can greatly reduce the number of free parameters from $n(n+1)M/2$ to $(n+M)R$. Throughout the paper, the CP-rank $R$ is allowed to diverge with $n$. In the CP decomposition of $\Theta$, $\alpha$ is the vertex latent position matrix with each row $\alpha_{i,\cdot}$ serving as the embedding of vertex $i$, and $\beta$ captures heterogeneity across different layers. Herein, we define the constraint sets for $\alpha$ and $\beta$ as $\Omega_{\alpha} = \{\alpha \in \mathbb{R}^{n \times R} : ||\alpha_{i,\cdot}|| \leq \sqrt{\log\frac{\xi}{1-\xi}}, \text{ for } i \in [n]\}$ and $\Omega_{\beta} = \{\beta \in \mathbb{R}^{M \times R} : ||\beta_{\cdot,r}|| = 1, r \in [R]\}$. Note that the constraint on $\beta$ is necessary for model identification, and detailed discussion will be presented shortly. The constraint set $\Omega_{\alpha} \times \Omega_{\beta}$ is sufficient to maintain the bounded condition of $\Theta$ since a general Hölder inequality yields that $|\theta_{i,j,m}| = |\mathcal{I} \times_1 \alpha_{i,\cdot}^T \times_2 \alpha_{j,\cdot}^T \times_3 \beta_{m,\cdot}^T| \leq ||\alpha_{i,\cdot}||||\alpha_{j,\cdot}||||\beta_{m,\cdot}||_{\infty} \leq \log\frac{\xi}{1-\xi}$. To conclude this paragraph, we remake that the parameter $\xi$ is introduced for theoretical purpose and it is not treated as a tuning parameter. One can choose $\xi$ sufficiently close to 1 in empirical studies so that the restriction on $\alpha$ will be alleviated.

We make several essential observations of the proposed TLSM. First and foremost, TLSM is flexible and general. It includes the celebrated MLSBM [34, 43, 35, 27, 26, 36, 22] as special case. Specif-ically, suppose the vertices comes form $K$ disjoint communities, the standard MLSBM assumes that the underlying network probability tensor $\mathcal{P} = \mathcal{B} \times_1 Z \times_2 Z$, where $\mathcal{B} \in \mathbb{R}^{K \times K \times M}$ is a semi-symmetric core probability tensor with $\mathcal{B}_{k_1,k_2,m} = \mathcal{B}_{k_2,k_1,m}$ for $k_1, k_2 \in [K]$ and $m \in [M]$, and $Z \in \{0,1\}^{n \times K}$ is the community membership matrix with $Z_{i,k} = 1$ if vertex $i$ comes from the $k$-th community and 0 otherwise. That is, the probability of any vertex pair to form an edge in a particular layer depends only on their community memberships. Equivalently, under the modified logit transformation (3), we have $\Theta = \widetilde{\mathcal{B}} \times_1 Z \times_2 Z$, where $\widetilde{\mathcal{B}}$ is the entry-wise transformation of $\mathcal{B}$ under (3). Taking $R$ to be the CP-rank of $\widetilde{\mathcal{B}}$, the CP-decomposition of $\widetilde{\mathcal{B}}$ then has the form

131  $\widetilde{\boldsymbol{\mathcal{B}}} = \boldsymbol{\mathcal{I}} \times_1 \boldsymbol{C} \times_2 \boldsymbol{C} \times_3 \boldsymbol{\beta}$ for some matrix $\boldsymbol{C} \in \mathbb{R}^{K \times R}$ and $\boldsymbol{\beta} \in \mathbb{R}^{M \times R}$ due to semi-symmetry.
132  This leads to the CP decomposition of $\boldsymbol{\Theta}$ has the form (4) with $\boldsymbol{\alpha} = \boldsymbol{ZC}$. It is clear that MLSBM
133  requires vertices within the same community are homogeneous and exchangeable, while TLSM
134  allows vertices to have different embeddings even when they are in the same community.

135  Second, TLSM is identifiable when both $\boldsymbol{\alpha}$ and $\boldsymbol{\beta}$ have full column ranks. When both $\boldsymbol{\alpha}$ and $\boldsymbol{\beta}$
136  have full column ranks, the Kruskal's k-ranks [25] of $\boldsymbol{\alpha}$ and $\boldsymbol{\beta}$ satisfy $k_{\boldsymbol{\alpha}} = k_{\boldsymbol{\beta}} = R$, then $\boldsymbol{\Theta}$ has
137  CP-rank $R$. Hence, $k_{\boldsymbol{\alpha}} + k_{\boldsymbol{\alpha}} + k_{\boldsymbol{\beta}} \geq 2R + 2$ as long as $R \geq 2$. By Theorem 1 of [40], the fixed
138  column $l_2$-norm constraint of $\boldsymbol{\beta}$ implies that the tensor factorization in (4) is unique up to column
139  permutations of $\boldsymbol{\alpha}$ and $\boldsymbol{\beta}$ and column sign flip of $\boldsymbol{\alpha}$. It is important to remark that the community
140  structure encoded in $\boldsymbol{\alpha}$ remains unchanged under any column permutation or sign flip.

141  Third, introducing a sparsity factor $s_n$ via a modified logit transformation into the TLSM is non-
142  trivial. We take a single-layer network as an example to illustrate the limitation of the standard
143  logit transformation in handling sparse network. Suppose a vanilla logit link is used to connect
144  the network underlying probability matrix $\boldsymbol{P}$ and its transformation $\boldsymbol{\Theta}$, and the latent space model
145  usually assumes that $\boldsymbol{\Theta} = \boldsymbol{\alpha}\boldsymbol{\alpha}^T$. A sparse network requires the entries of $\boldsymbol{\Theta}$ diverge to negative
146  infinite due to the small magnitude of edge probability, which leads to unstable estimation of $\boldsymbol{\alpha}$ in
147  numerical experiments. Moreover, this may conflict with the assumption that vertices within the same
148  community tend to be close in the embedding space and their inner product is likely to be positive.
149  These difficulties can be naturally circumvented when an appropriate $s_n$ is chosen in (3).

## 2.2  Regularized likelihood

Given a network adjacency tensor $\boldsymbol{\mathcal{A}}$ and number of communities $K$, our goal is to estimate the
multi-layer network embedding $(\boldsymbol{\alpha}, \boldsymbol{\beta})$ and conduct community detection on the vertices. Throughout
this paper, we assume the number of potential communities $K$ is given and may diverge with $n$. Under
the TLSM framework, with slight abuse of notation, we denote the average negative log-likelihood
function of the multi-layer network $\mathcal{G}$ is $\mathcal{L}(\boldsymbol{\alpha}, \boldsymbol{\beta}; \boldsymbol{\mathcal{A}}) = \mathcal{L}(\boldsymbol{\Theta}; \boldsymbol{\mathcal{A}})$ with

$$\mathcal{L}(\boldsymbol{\Theta}; \boldsymbol{\mathcal{A}}) = \frac{1}{\varphi(n, M)} \sum_{m=1}^{M} \sum_{i \leq j} L(\theta_{i,j,m}; a_{i,j,m}),$$

151  where $\varphi(n, M) = \frac{1}{2}n(n+1)M$ is the number of potential edges, and $L(\theta; a) = \log\left(1 + \frac{s_n}{1 - s_n + e^{-\theta}}\right) -$
152  $a \log\left(\frac{s_n}{1 - s_n + e^{-\theta}}\right)$ is a negative log-density of a Bernoulli random variable $a$. We now introduce a
153  novel regularization term to detect the potential communities in $\mathcal{G}$,

$$J(\boldsymbol{\alpha}) = \min_{\boldsymbol{Z} \in \Gamma, \boldsymbol{C} \in \mathbb{R}^{K \times R}} \frac{1}{n} \|\boldsymbol{\alpha} - \boldsymbol{ZC}\|_F^2, \tag{5}$$

154  where $\boldsymbol{C}$ encodes the vertex embedding centers and $\Gamma \subset \{0, 1\}^{n \times K}$ is the set of all possible
155  community membership matrices; that is, for any $\boldsymbol{Z} \in \Gamma$, each row of $\boldsymbol{Z}$ consists of only one 1
156  indicating the community membership and all others entries being 0. This leads to the proposed
157  regularized cost function,

$$\mathcal{L}_\lambda(\boldsymbol{\alpha}, \boldsymbol{\beta}; \boldsymbol{\mathcal{A}}) = \mathcal{L}(\boldsymbol{\alpha}, \boldsymbol{\beta}; \boldsymbol{\mathcal{A}}) + \lambda_n J(\boldsymbol{\alpha}), \tag{6}$$

158  where $\lambda_n$ is a positive tuning parameter that strikes the balance between network estimation and
159  community detection in the cost function. It is clear that the embeddings of vertices with similar
160  linking pattern will be pushed towards the same center, and thus close to each other in the ambient
161  space, leading to the desired community structure in $\mathcal{G}$.

## 2.3  Projected gradient descent algorithm

163  We develop a scalable projected gradient descent (PGD) algorithm to optimize the penalized cost
164  function (6), which is highly non-convex and can be solved only locally. PGD, which alternatively
165  conducts gradient step and projection step, is one of the most popular and computationally fast
166  algorithm in tackling non-convex optimization problem [7, 33, 47, 9].

167  To compute the gradients of $\boldsymbol{\alpha}$ and $\boldsymbol{\beta}$, we introduce the following notations. Define $\boldsymbol{\mathcal{T}} \in \mathbb{R}^{n \times n \times M}$
168  with entries $\boldsymbol{\mathcal{T}}_{i,j,m} = \frac{\exp(-\theta_{i,j,m})}{1 - s_n + \exp(-\theta_{i,j,m})}(p_{i,j,m} - a_{i,j,m})$, and $\boldsymbol{X}_{\boldsymbol{\mathcal{T}}(2,3)}^{\boldsymbol{\alpha}, \boldsymbol{\beta}} \in \mathbb{R}^{n \times R}$ whose $i$-th row

consists of the diagonal elements of the slice $(\mathcal{T} \times_2 \boldsymbol{\alpha}^T \times_3 \boldsymbol{\beta}^T)_{i,\cdot,\cdot}$. That is, $\boldsymbol{X}^{\boldsymbol{\alpha},\boldsymbol{\beta}}_{\mathcal{T}(2,3)}(i,r) =$ $(\mathcal{T} \times_2 \boldsymbol{\alpha}^T \times_3 \boldsymbol{\beta}^T)_{i,r,r}$. Similarly, we define $\boldsymbol{X}^{\boldsymbol{\alpha},\boldsymbol{\alpha}}_{\mathcal{T}(1,2)} \in \mathbb{R}^{R \times M}$, $\boldsymbol{X}^{\boldsymbol{\beta}}_{\mathcal{T}(3)} \in \mathbb{R}^{n \times R}$, and $\boldsymbol{X}_{\mathcal{T}(1,2)} \in$ $\mathbb{R}^{n \times M}$, such that $\boldsymbol{X}^{\boldsymbol{\alpha},\boldsymbol{\alpha}}_{\mathcal{T}(1,2)}(r,m) = (\mathcal{T} \times_1 \boldsymbol{\alpha}^T \times_2 \boldsymbol{\alpha}^T)_{r,r,m}$, $\boldsymbol{X}^{\boldsymbol{\beta}}_{\mathcal{T}(3)}(i,r) = (\mathcal{T} \times_3 \boldsymbol{\beta}^T)_{i,i,r}$, and $\boldsymbol{X}_{\mathcal{T}(1,2)}(i,m) = \mathcal{T}_{i,i,m}$. Consequently, when the vertex membership matrix $\boldsymbol{Z}$ and the community center matrix $\boldsymbol{C}$ are fixed, we can derive the gradients of $\mathcal{L}_\lambda(\boldsymbol{\alpha},\boldsymbol{\beta};\mathcal{A})$ with respect to $\boldsymbol{\alpha}$ and $\boldsymbol{\beta}$, as

$$\frac{1}{\varphi(n,M)}\left(\boldsymbol{X}^{\boldsymbol{\alpha},\boldsymbol{\beta}}_{\mathcal{T}(2,3)} + \boldsymbol{X}^{\boldsymbol{\beta}}_{\mathcal{T}(3)} * \boldsymbol{\alpha}\right) + 2\lambda_n(\boldsymbol{\alpha} - \boldsymbol{Z}\boldsymbol{C}) \text{ and } \frac{1}{2\varphi(n,M)}\left((\boldsymbol{X}^{\boldsymbol{\alpha},\boldsymbol{\alpha}}_{\mathcal{T}(1,2)})^T + \boldsymbol{X}^T_{\mathcal{T}(1,2)}(\boldsymbol{\alpha} * \boldsymbol{\alpha})\right),$$

respectively. Herein, * denotes the Hadamard product (entry-wise product) between two matrices.

Let $(\tilde{\boldsymbol{\alpha}}, \tilde{\boldsymbol{\beta}})$ denote the solution given by one-step gradient descent, we then project $(\tilde{\boldsymbol{\alpha}}, \tilde{\boldsymbol{\beta}})$ onto $\Omega_{\boldsymbol{\alpha}} \times \Omega_{\boldsymbol{\beta}}$ in the following steps.

*Step 1.* Multiply the $r$-th column of $\tilde{\boldsymbol{\alpha}}_{\cdot,r}$ by $||\tilde{\boldsymbol{\beta}}_{\cdot,r}||^{1/2}$ for $r \in [R]$. Denote the resultant matrix as $\tilde{\boldsymbol{\alpha}}'$.

*Step 2.* Regularize each row of $\boldsymbol{\alpha}$ as $\boldsymbol{\alpha}_{i,\cdot} = \tilde{\boldsymbol{\alpha}}'_{i,\cdot} \min\{\sqrt{\log\frac{\xi}{1-\xi}}, ||\tilde{\boldsymbol{\alpha}}'_{i,\cdot}||\}/||\tilde{\boldsymbol{\alpha}}'_{i,\cdot}||$, for $i \in [n]$.

*Step 3.* Normalize the columns of $\boldsymbol{\beta}$ as $\boldsymbol{\beta}_{\cdot,r} = \tilde{\boldsymbol{\beta}}_{\cdot,r}/||\tilde{\boldsymbol{\beta}}_{\cdot,r}||$, for $r \in [R]$.

Next, when $(\boldsymbol{\alpha}, \boldsymbol{\beta})$ are given, we apply a $(1+\delta)$-approximation K-means algorithm on $\tilde{\boldsymbol{\alpha}}$ to update the vertex community membership matrix $\boldsymbol{Z}$ and community center matrix $\boldsymbol{C}$.

The above steps will be alternatively conducted until convergence or reaching the maximum number of iterations. We further summarized the developed alternative updated scheme in Algorithm 1 in Appendix A of the supplementary materials

Several remarks on the algorithm are in order. First, Algorithm 1 can only be guaranteed to converge to a stationary point but not any local minimizer. We hence employ a transformed higher order orthogonal iteration (HOOI) algorithm for warm initialization in all the numerical experiments in Section 4 and 5. Specifically, given a user-specific value $\tau$, we define $\widetilde{\boldsymbol{\Theta}}$ to mimic the magnitude of $\boldsymbol{\Theta}$ such that $\widetilde{\boldsymbol{\Theta}}_{i,j,m} = -\tau$ if $a_{i,j,m} = 0$ and $\widetilde{\boldsymbol{\Theta}}_{i,j,m} = \tau$ otherwise. A standard HOOI algorithm [11] is applied to $\widetilde{\boldsymbol{\Theta}}$ to obtain $\boldsymbol{\alpha}^{(0)}$ and $\boldsymbol{\beta}^{(0)}$. We set $\tau = 100$ in all the numerical experiments. Second, the sparsity factor $s_n$ is an intrinsic quantity of the multi-layer network data, and it should be estimated from the network directly. Note that the minimal and maximal probabilities for any vertex pair to form an edge in any layer are $p_{\min} = (1-\xi)s_n$ and $p_{\max} = \xi s_n$, respectively. Interestingly, $p_{\min} + p_{\max} = s_n$, which does not depend on $\xi$ any more. Therefore, we propose to estimate $s_n$ as

$$\hat{s}_n = \min_{i \in [n]} \frac{1}{nM} \sum_{m=1}^{M} \sum_{j=1}^{n} a_{i,j,m} + \max_{i \in [n]} \frac{1}{nM} \sum_{m=1}^{M} \sum_{j=1}^{n} a_{i,j,m}, \tag{7}$$

which is the sum of the minimal and maximal frequencies of a vertex to form edges with all other vertices in all layers. Third, to optimally choose $\lambda_n$, we extend the network cross-validation by edge sampling scheme in [30] to multi-layer networks. The detailed tuning procedure is relegated to Appendix B in the supplementary materials.

# 3 Asymptotic theory

## 3.1 Consistency in estimating $\Theta^*$

Let $\Omega = \{\boldsymbol{\Theta} = \mathcal{I} \times_1 \boldsymbol{\alpha} \times_2 \boldsymbol{\alpha} \times_3 \boldsymbol{\beta} : \boldsymbol{\alpha} \in \Omega_{\boldsymbol{\alpha}}, \boldsymbol{\beta} \in \Omega_{\boldsymbol{\beta}}\}$ be the parameter space of the problem and $\boldsymbol{\Theta}^* = \mathcal{I} \times_1 \boldsymbol{\alpha}^* \times_2 \boldsymbol{\alpha}^* \times_3 \boldsymbol{\beta}^*$ be the true underlying transformed network probability tensor. Denote $KL(\boldsymbol{\Theta}^*||\boldsymbol{\Theta}) = \varphi^{-1}(n,M) \sum_{m=1}^{M} \sum_{i \leq j} E\left(L(\theta_{i,j,m}; a_{i,j,m}) - L(\theta^*_{i,j,m}; a_{i,j,m})\right)$ be the averaged Kullback–Leibler divergence of the network generation distributions parametrized by $\boldsymbol{\Theta}^*$ and $\boldsymbol{\Theta}$, for any $\boldsymbol{\Theta} \in \Omega$. The following large deviation inequality is derived to quantify the behavior of $\mathcal{L}_\lambda(\boldsymbol{\Theta}; \mathcal{A})$ for any $\boldsymbol{\Theta}$ in the neighborhood of $\boldsymbol{\Theta}^*$ defined by $KL(\boldsymbol{\Theta}^*||\boldsymbol{\Theta})$.

**Proposition 1.** *Suppose $\lambda_n J(\boldsymbol{\alpha}^*) \leq \epsilon_n$, and $(n+M)R\varphi^{-1}(n,M)\epsilon_n^{-1}\log(\epsilon_n^{-1/2}) \leq c_1$ for some constant $c_1$. Then with probability at lease $1 - 2\exp\left(-\frac{\varphi(n,M)\epsilon_n}{156\frac{\xi}{1-\xi} + 28\log 2}\right)$, we have*

$$\mathcal{L}_\lambda(\boldsymbol{\Theta}^*; \mathcal{A}) \leq \inf_{\{\boldsymbol{\Theta} \in \Omega | KL(\boldsymbol{\Theta}^*||\boldsymbol{\Theta}) \geq 4\epsilon_n\}} \mathcal{L}_\lambda(\boldsymbol{\Theta}; \mathcal{A}) - \epsilon_n.$$

Proposition 1 basically states that any estimators with sufficiently small objective value should be close enough to $\boldsymbol{\Theta}^*$ in terms of $KL(\boldsymbol{\Theta}^*||\boldsymbol{\Theta})$. We next study the asymptotic behavior of these estimators more precisely. Let $(\hat{\boldsymbol{\alpha}}, \hat{\boldsymbol{\beta}}) \in \Omega_{\boldsymbol{\alpha}} \times \Omega_{\boldsymbol{\beta}}$ be any estimator of $(\boldsymbol{\alpha}^*, \boldsymbol{\beta}^*)$ such that

$$\mathcal{L}_\lambda(\hat{\boldsymbol{\alpha}}, \hat{\boldsymbol{\beta}}; \mathcal{A}) \leq \mathcal{L}_\lambda(\boldsymbol{\alpha}^*, \boldsymbol{\beta}^*; \mathcal{A}) + \epsilon_n, \tag{8}$$

and denote $\widehat{\boldsymbol{\Theta}} = \mathcal{I} \times_1 \hat{\boldsymbol{\alpha}} \times_2 \hat{\boldsymbol{\alpha}} \times_3 \hat{\boldsymbol{\beta}}$. we have the following theorem.

**Theorem 1.** *Under the condition of Proposition 1, if $(\hat{\boldsymbol{\alpha}}, \hat{\boldsymbol{\beta}})$ satisfies (8), then with probability at least $1 - 2\exp\left(-\frac{\varphi(n,M)\epsilon_n}{156\frac{\xi}{1-\xi}+28\log 2}\right)$, we have*

$$\frac{1}{n\sqrt{M}}\|\widehat{\boldsymbol{\Theta}} - \boldsymbol{\Theta}^*\|_F \leq \frac{4\sqrt{2}\sqrt{\epsilon_n}}{(1-\xi)\sqrt{\xi}s_n}.$$

The condition that $\lambda_n J(\boldsymbol{\Theta}^*) \leq \epsilon_n$ in Proposition 1 is mild. It implies that the true embeddings of vertices within the same community are close to one another. We remark that $\lambda_n J(\boldsymbol{\Theta}^*)$ exactly equals to zero under the MLSBM discussed in Section 2.2. The condition that $(n+M)R\varphi^{-1}(n,M)\epsilon_n^{-1}\log(\epsilon_n^{-1/2})$ vanishes with $n$ is also mild. When $R = O(1)$, we can take any $\epsilon_n$ such that $\epsilon_n \gg \frac{\log n}{n\min\{n,M\}}$. Consequently, to ensure $\widehat{\boldsymbol{\Theta}}$ converges to $\boldsymbol{\Theta}^*$, Theorem 1 implies the smallest sparsity factor one can take is $s_n \gg \epsilon_n \gg \frac{\log n}{n\min\{n,M\}}$, which means that the average degree of a vertex in any particular layer can be as small as $ns_n$. We remark that a common assumption $M = O(n)$ that appears in literature, such as [27] and [22], is not necessary in our theory. If we further assume $M = O(n)$, we find that the average degree of a vertex in any layer under the proposed TLSM set up can be smaller than that in [27] by a factor $(M\log n)^{-1/2}$ and in [22] by a factor $(\log n)^{-3}$, showing that our theoretical result accommodates sparser multi-layer networks.

## 3.2 Consistency in community detection

We now turn to establish the consistency of community detection in multi-layer network $\mathcal{G}$. Let $\psi^* : [n] \longrightarrow [K]$ be the true community assignment function such that $\psi^* = \arg\min_\psi \min_{C_1,\ldots,C_K} \sum_{i=1}^n \|\boldsymbol{\alpha}_i^* - C_{\psi_i}\|^2$, and then the community detection error of any estimated community assignment function $\hat{\psi}$ can be evaluated by the minimum scaled Hamming distance between $\hat{\psi}$ and $\psi^*$ under permutations, which is defined as

$$\text{err}(\psi^*, \hat{\psi}) = \min_{\pi \in S_K} \frac{1}{n}\sum_{i=1}^n \mathbf{1}\{\psi_i^* \neq \pi(\hat{\psi}_i)\}, \tag{9}$$

where $\mathbf{1}\{\cdot\}$ is the indicator function and $S_K$ is the symmetric group of degree $K$. Such a scaled or unscaled Hamming distance has become a popular metric in quantifying the performance of community detection [21, 22].

Denote $N_k^* = \{i : \psi_i^* = k\}$ be the $k$-th true underlying community whose cardinality is $n_k$. Let $\boldsymbol{C}^* \in \mathbb{R}^{K \times R}$ be the true underlying community centers of the network embedding with $\boldsymbol{C}_{k.}^* = \frac{1}{n_k}\sum_{\psi_i^*=k}\boldsymbol{\alpha}_{i.}^*$, and let $\boldsymbol{\mathcal{B}}^* = \mathcal{I} \times_1 \boldsymbol{C}^* \times_2 \boldsymbol{C}^* \times_3 \boldsymbol{\beta}^*$. The following assumptions are made to ensure that communities within the multi-layer networks are asymptotically identifiable.

**Assumption A.** *Assume the difference between any two distinct horizontal slides of $\boldsymbol{\mathcal{B}}^*$ satisfies that*

$$\min_{k,k' \in [K], k \neq k'} \frac{1}{\sqrt{KM}}\|\boldsymbol{\mathcal{B}}_{k,:,:}^* - \boldsymbol{\mathcal{B}}_{k',:,:}^*\|_F \geq \gamma_n,$$

*where $\gamma_n > 0$ may vanish with $n$.*

**Assumption B.** *Assume the tuning parameter $\lambda_n$ satisfies that*

$$\lambda_n \epsilon_n s_n^{-2}(\log s_n^{-1})^{-1} \geq c_2,$$

*for an absolute constant $c_2$ that does not depend on any model parameter.*

**Assumption C.** *Denote $n_{\min} = \min_{k \in [K]} n_k$ as the minimal community size. Assume*

$$\frac{\gamma_n n_{\min}\sqrt{K}}{n} \geq c_\xi\sqrt{\frac{\epsilon_n}{s_n}},$$

*where $c_\xi = \frac{4\sqrt{2}}{(1-\xi)\sqrt{\xi}} + c_3\sqrt{\frac{(1+\delta)\min\{M,R\}}{M}}$ and $c_3$ is a constant that depends on $\xi$ only.*

Assumption A is the minimal community separation requirement, and similar assumption has been employed in [27] with a constant $\gamma_n$. Together with the condition $\lambda_n J(\boldsymbol{\alpha}^*) \le \epsilon_n$ in Proposition 1, Assumption B gives a feasible interval for $\lambda_n$. Assumption C allows for unbalanced communities with vanishing $n_{\min}/n$ if the network is not too sparse. Note that $c_\xi$ can be further bounded by $\frac{4\sqrt{2}}{(1-\xi)\sqrt{\xi}} + c_3\sqrt{1+\delta}$, and the first term of $c_\xi$ will dominate the second term if $R = o(M)$.

**Theorem 2.** *Suppose all the assumptions in Theorem 1 as well as Assumptions A, B and C are satisfied, it holds true that*

$$ err(\psi^*, \hat{\psi}) \le \frac{c_\xi^2 n \epsilon_n}{n_{\min} K \gamma_n^2 s_n}, $$

*with probability at least* $1 - \frac{1}{n^2} - 2\exp\left(-\frac{\varphi(n,M)\epsilon_n}{156\frac{\xi}{1-\xi} + 28\log 2}\right)$.

Theorem 2 assures that the community structure in a multi-layer network can be consistently recovered by the proposed TLSM. As a theoretical example, we consider a sparse case with $s_n = \frac{(\log n)^{1+\tau_1}}{n \min\{n,M\}}$, where $0 < \tau_1 < 1$, $n_{\max} = O(n_{\min})$, $\frac{1}{\sqrt{n}}\|\boldsymbol{\alpha}^* - \boldsymbol{Z}^*\boldsymbol{C}^*\|_F \le (\log n)^{-3/2}$, and both $\gamma_n$, $R$ and $K$ are of constant orders. With $\lambda_n = \frac{(\log n)^{2+2\tau_1}}{n \min\{n,M\}}$, Theorems 1 and 2 imply that $\epsilon_n = \frac{(\log n)^{1+\tau_2}}{n \min\{n,M\}}$ with $0 < \tau_2 < \tau_1$ and $err(\psi^*, \hat{\psi}) = o_p(1)$.

# 4 Numerical experiments

In this section, we evaluate the numerical performance of the proposed TLSM in a variety of synthetic as well as real-life multi-layer networks, compare it against four competitors in literature, including the mean adjacency spectral embeddings (MASE; 16), least square estimation (LSE; 27), Tucker decomposition with HOSVD initialization (HOSVD-Tucker; 22), and spectral kernel (SPECK; 35), and conduct some ablation studies. The implementations of LSE and SPECK are available at the authors' personal websites, HOSVD-Tucker is implemented in the routine "tucker" of the Python package "tensorly", and TLSM and MASE are implemented in Python by ourselves.

## 4.1 Synthetic networks

The multi-layer network $\mathcal{A} = (a_{i,j,m}) \in \{0,1\}^{n \times n \times M}$ is generated as follows. First, we randomly select $K = 4$ elements uniformly from $\{2.5*(b_1, b_2, \ldots, b_R) : b_r \in \{-1,1\}, r \in [R]\}$ as community centers, which are denoted as $\boldsymbol{c}_k$, $k \in [K]$. Second, the latent space embedding of vertex $i$ is generated as $\boldsymbol{\alpha}_i = \boldsymbol{c}_{\psi_i} + \boldsymbol{e}_i$ with $\boldsymbol{e}_i \sim N(\boldsymbol{0}_R, 1.5 * I_R)$, and $\psi_i \in [K]$ are independently drawn from the multinomial distribution Multi$(1; \frac{1}{K}\boldsymbol{1}_K)$. Third, we generate $\boldsymbol{\beta} = [\boldsymbol{\beta}_1, \ldots, \boldsymbol{\beta}_M]^T$ with $\boldsymbol{\beta}_{m,r}$ being independent standard normal random varibeles, for $m \in [M]$ and $r \in [R]$. We then rescale the column norms of $\boldsymbol{\beta}$ to be 1 for model identifiability. Finally, we generate $\mathcal{A}$ according to the proposed TLSM with $s_n = 0.1$. For the sake of fair comparisons, the embedding dimension $R$ is set as $K$ in all scenarios. We aim to illustrate the community detection performance of all methods as the number of vertices and number of layers increase. To this end, we consider $(n, M) \in \{200, 400, 600, 800\} \times \{5, 10, 15, 20\}$. The averaged hamming errors and their standard errors over 50 independent experiments of all methods are reported in Table 1.

It is evident that TLSM consistently outperforms its competitors, and the performances of LSE and HOSVD-Tucker are better than those of MASE and SPECK. This is expected since TLSM, LSE and HOSVD-Tucker work on the multi-layer network adjacency tensor directly, while MASE and SPECK are matrix aggregation methods that suffer form information loss. Furthermore, as the number of vertices and number of layers increase, the community detection errors of all methods decrease rapidly. Notably, TLSM and LSE converge faster than the other methods, and attain stable performance even for relatively small $n$ and $M$. Additional simulation studies for various network sparsity and unbalanced community sizes are relegated to Appendix C in the supplementary materials.

## 4.2 Real-life networks

We also apply the proposed TLSM method to analyze three real-life multi-layer networks, including a social network in the department of Computer Science at Aarhus University (AUCS) [38], a yeast

Table 1: The averaged hamming errors of various methods with their standard errors in Scenario I. The best performer in each case is bold-faced.

| $n$ | $M$ | TLSM | LSE | MASE | HOSVD-Tucker | SPECK |
|---|---|---|---|---|---|---|
| 200 | 5 | **0.1180**(0.0147) | 0.1405(0.0118) | 0.5086(0.0136) | 0.1623(0.0126) | 0.4254(0.0138) |
| | 10 | **0.0585**(0.0046) | 0.0751(0.0050) | 0.4949(0.0131) | 0.1148(0.0106) | 0.2996(0.0141) |
| | 15 | **0.0551**(0.0067) | 0.0593(0.0045) | 0.4910(0.0176) | 0.1040(0.0115) | 0.2505(0.0142) |
| | 20 | **0.0510**(0.0037) | 0.0588(0.0043) | 0.4977(0.0161) | 0.1023(0.0110) | 0.1942(0.0156) |
| 400 | 5 | **0.0653**(0.0066) | 0.1019(0.0087) | 0.3845(0.0193) | 0.1220(0.0106) | 0.3766(0.0195) |
| | 10 | **0.0608**(0.0063) | 0.0636(0.0037) | 0.3859(0.0160) | 0.1012(0.0092) | 0.2244(0.0191) |
| | 15 | **0.0511**(0.0031) | 0.0595(0.0036) | 0.3844(0.0221) | 0.0787(0.0051) | 0.1490(0.0123) |
| | 20 | **0.0536**(0.0047) | 0.0551(0.0036) | 0.3985(0.0185) | 0.0795(0.0063) | 0.1409(0.0131) |
| 600 | 5 | **0.0607**(0.0029) | 0.0909(0.0040) | 0.3665(0.0186) | 0.1221(0.0108) | 0.3038(0.0193) |
| | 10 | **0.0567**(0.0029) | 0.0688(0.0031) | 0.3726(0.0179) | 0.1003(0.0081) | 0.1651(0.0127) |
| | 15 | **0.0558**(0.0027) | 0.0630(0.0030) | 0.3803(0.0167) | 0.0918(0.0076) | 0.1231(0.0076) |
| | 20 | **0.0548**(0.0028) | 0.0586(0.0029) | 0.3814(0.0185) | 0.0883(0.0078) | 0.1150(0.0088) |
| 800 | 5 | **0.0556**(0.0056) | 0.0768(0.0055) | 0.3012(0.0194) | 0.1003(0.0103) | 0.2733(0.0171) |
| | 10 | **0.0560**(0.0063) | 0.0583(0.0034) | 0.3004(0.0177) | 0.0788(0.0065) | 0.1424(0.0127) |
| | 15 | **0.0498**(0.0030) | 0.0539(0.0033) | 0.3179(0.0195) | 0.0812(0.0068) | 0.1146(0.0098) |
| | 20 | **0.0485**(0.0031) | 0.0516(0.0032) | 0.3184(0.0218) | 0.0803(0.0075) | 0.0979(0.0078) |

Saccharomyces cerevisiae gene co-expression (YSCGC) network [44], and a worldwide agriculture trading network (WAT) [10]. Specifically, we conduct community detection on the first two networks whose vertex community memberships are available, and carry out a link prediction task on the third network whose vertex community memberships are unavailable.

The AUCS dataset is publicly available at `http://multilayer.it.uu.se/datasets.html`, and it is a $61 \times 61 \times 5$ multi-layer network that records pairwise relationships of 5 types among 61 persons in AUCS, including current working relationships, repeated leisure activities, regularly eating lunch together, co-authorship of a publication, and friendship on Facebook. Since 54 persons in the dataset come from 7 research groups and the other 7 persons do not belong to any group, the dataset consists of 8 communities corresponding to 7 research groups and an outlier community. Applying TLSM and its competitors to the dataset, the number of misclassified vertices by TLSM, LSE, MASE, HOSVD-Tucker and SPECK, are 8, 21, 19, 23, 18, respectively. Clearly, TLSM significantly outperforms its competitors by at least reducing $16.39\%$ of community detection error.

The YSCGC dataset is publicly available at `https://www.ncbi.nlm.nih.gov/pmc/articles/PMC156590/`, and contains 205 genes of 4 functional categories, including protein metabolism and modification, carbohydrate metabolism and catabolism, nucleobase, nucleoside, nucleotide and nucleic acide metabolism, as well as transportation. We regard these four functional category labels as the community memberships of the genes. Further, the gene expression responses are measured by 20 systematic perturbations with varying genetic and environmental conditions in 4 replicated hybridizations. We thus constructed a gene co-expression network $\mathcal{A} = (a_{i,j,m}) \in \mathbb{R}^{205 \times 205 \times 4}$ based on the similarities of their expressions, where each layer represents one replicated hybridization. Specifically, the similarity between genes $i$ and $j$ in the $m$-th replication is measured by $w_{i,j,m} = \exp\left(-\|\boldsymbol{x}_i^{(m)} - \boldsymbol{x}_j^{(m)}\|\right)$, where $\boldsymbol{x}_i^{(m)} \in \mathbb{R}^{20}$ contains the expression levels of 20 perturbations in the $m$-th replicated hybridization for $i \in [205]$ and $m \in [4]$. The binary value $a_{i,j,m}$ is obtained by thresholding $w_{i,j,m}$ with the thresholding value being the $60\%$ quantile of all elements in $\{w_{i,j,m} : i \le j \in [205], m \in [4]\}$. Applying TLSM and its competitors to this dataset, the number of misclassified vertices by TLSM, LSE, MASE, HOSVD-Tucker and SPECK, are 6, 9, 12, 48, 13, respectively. TLSM again outperforms its competitors in this YSCGC dataset.

The WAT dataset is publicly available at `http://www.fao.org`, and includes 364 agriculture product trading relationships among 214 countries in 2010. To process the data, we extract 130 major countries whose average degrees are greater than 9 from the 32 densest connected agriculture product trading relations, leading to a $130 \times 130 \times 32$ multi-layer network. Investigating the eigen-structure of the mode-1 matricization of the network adjacency tensor, we identify an elbow point [20] at the 7th largest eigen-value, suggesting there are 6 potential communities among the countries, and thus we set $K = 6$. The corresponding eigen-value plot is attached in Appendex D of the supplementary materials. We then randomly selected $80\%$ of the entries of the adjacency tensor as the training set, and conduct link prediction on the remaining $20\%$ of the entries. Specifically, we employ TLSM

and the adaptations of its competitors to estimate the network expected tensor $\mathcal{P}$ and generate estimations for the missing entries by independent Bernoulli random variables accordingly. The averaged link prediction accuracy of TLSM, LSE, MASE, HOSVD-Tucker and SPECK over 50 independent replications are 79.60%, 76.66%, 75.96%, 77.78% and 79.08%, respectively, where the link prediction accuracy is defined as the percentile of the correctly predicted entries. Clearly, all 5 methods are comparative in terms of link prediction, while TLSM still deliver highest averaged link prediction accuracy.

### 4.3 Ablation studies

In this subsection, we carry out some ablation studies on two novel components of the proposed method, namely the sparsity factor $s_n$ and the community-inducing regularizer $J(\boldsymbol{\alpha})$. To study the effectiveness of $s_n$, we generate a $300 \times 300 \times 5$ multi-layer network with 3 communities and the true network sparsity $s_n = 0.3$. The blue curve in the left panel of Figure 1 shows the average Hamming error of 50 independent replications given by the proposed method when employing $\hat{s}_n \in \{0.05i : i \in [20]\}$ in the optimization algorithm, and the red line indicates the averaged Hamming error of the proposed method with $\hat{s}_n$ estimated via the proposed data-adapted estimation scheme. It is clear that the Hamming error at $s_n = 1$ is much larger than that when $s_n$ is close to 0.3, showing the advantages of the modified logit transformation by $s_n$ over the standard logit transformation when the network indeed reveals sparse pattern. Moreover, we observe that the red line is even lower than the minimum Hamming error in the blue curve. This further confirms the effectiveness of the proposed data-adapted estimation scheme for estimating $s_n$.

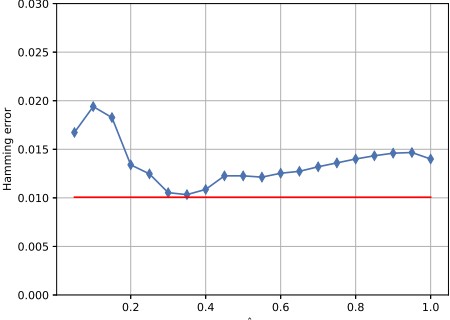 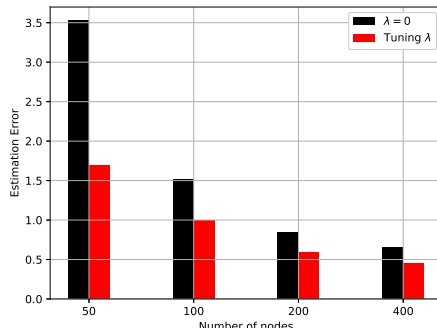

Figure 1: Ablation studies on $s_n$ (left) and community-inducing regularizer (right).

To study the effectiveness of the community-inducing regularizer in the proposed objective function, we generate an $n \times n \times 5$ multi-layer network with 2 communities, for $n \in \{50, 100, 200, 400\}$. In the right panel of Figure 1, the black pillars indicate the network estimation error $\frac{1}{n\sqrt{5}}\|\widehat{\boldsymbol{\Theta}} - \boldsymbol{\Theta}^*\|_F$ given by the proposed method with $\lambda_n = 0$ which corresponds to the absence of $J(\boldsymbol{\alpha})$, while the red ones indicate the counterparts given by the proposed method with $\lambda_n$ is selected by network cross-validation. There is a clear improvement when the community-inducing regularizer is enforced in all scenarios, particularly for small $n$. This showcases the helpfulness of the community-inducing regularizer in detecting network community structure.

## 5 Conclusions

In this paper, we propose a novel tensor-based latent space model for community detection in multi-layer networks. The model embeds vertices into a low-dimensional latent space and views the community structure from an network embedding perspective, so that heterogeneous structures in different network layers can be properly integrated. The proposed model is formulated as a regularization framework, which conducts multi-layer network estimation and community detection simultaneously. The advantages of the proposed method are supported by extensive numerical experiments and theoretical results. Particularly, the asymptotic consistencies of the proposed method are established in terms of both multi-layer network estimation and community detection, even for relatively sparse networks.

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
