# OpenReview forum: "Structure-Preserving Embedding of Multi-layer Networks"
_NeurIPS.cc/2022/Conference — NeurIPS 2022 Submitted_

### Official Review · Reviewer_q91p · 2022-07-11

**Rating:** 5
**Confidence:** 4
**Soundness:** 3 good
**Presentation:** 2 fair
**Contribution:** 2 fair

**Summary:**

In this paper, the authors propose TLSM, a novel tensor-based latent space model for community detection in multi-layer networks. TLSM integrates the heterogenous network structure in different layers by embedding nodes into a low-dimensional space with the aim of nodes within the same community should have closer embeddings. Further, TLSM utilizes a regularized framework consisting of the average negative log-likelihood function of the multi-layer network and the clustering regularizer, to estimate the multi-layer network and conduct community detection simultaneously. The authors also provide theoretical analysis regarding the asymptotic consistencies of TLSM of both the multi-layer network and community detection.

**Questions:**

1. It seems that TLSM mainly leverages the tensor CP decomposition to integrate the heterogeneous structure of the multi-layer networks. The reviewer wonders if the integration would be better if TLSM incorporates random walks between different layers.

2. The computational complexity of TLSM is not mentioned in the paper, and thus it would be great if the author could discuss further TLSM's complexity.

**Limitations:**

Authors did not mention the Limitations and societal impacts. The reviewer thinks the limitations of the proposed model are mainly the optimality of learned parameters and the model's performance.

**Strengths And Weaknesses:**

Strengths:

1. TLSM is a flexible and general framework that contains many multi-layer network generative models such as the multi-layer stochastic block model.

2. TLSM estimates the multi-layer network and performs community detection simultaneously by adding a clustering penalty to the multi-layer network likelihood function.

3. TLSM analyzes the asymptotic consistencies in terms of both the multi-layer network and the community detection.

Weaknesses:

1. TLSM applies projected gradient descent to optimize the regularized likelihood, which can only guarantee achieving a local optimum. The authors mentioned employing a transformed higher order orthogonal iteration (HOOI) algorithm for warm initialization, but it would be great if the authors could discuss it further in detail.

2. TLSM outperforms baseline methods in the synthetic networks, while does not significantly outperform baselines in the real-world networks. The reviewer wonders if the authors could provide more details about the real-world experiments.

3. Although TLSM is flexible and general, the techniques it used are not novel.

---

> ### Author Response · Authors · 2022-08-02
> **Responds to Reviewer q91p**
>
> Thank you very much for your valuable comments, which will significantly help us improve the quality of the manuscript. In what follows, we address all your comments point by point with our best efforts.
>
> Response to Weakness 1: In fact, obtaining an optimal low-rank approximation of a tensor is computationally intractable [1]. Usually, algorithms with random initializations suffer from non-informative local minimals. To avoid this, it is a common practice to employ a warm initialization before updating the tensor parameter estimation. In this paper, we employ HOOI as the initialization algorithm due to its low computational cost and better performance compared with HOSVD. In literature, HOOI has been widely employed for warm initialization in high-order network data analysis [2].
>
> Response to Weakness 2:  In the real applications, the proposed method actually yields some substantial improvement over the baseline approaches in term of community detection, while its performances is still slightly better than the baseline approaches in term of link prediction. In the AUCS dataset, the improvement of TLSM over the baseline methods in term of community detection errors ranges from 55.5\% to 65.2\%. In the YSCGC dataset, the improvement of TLSM ranges from 33.3\% to 87.5\%. As for the WAT dataset, the link prediction accuracy of TLSM is still slightly better than the baseline methods, which may be further improved when the current community-inducing penalty is replaced by the below-mentioned ridge penalty.
>
> Response to Weakness 3: The proposed method builds upon tensor decomposition to incorporate the heterogeneity among layers. This is in sharp contrast to most existing methods that aggregate network layers into a single one, which may lead to information loss or sub-optimal theoretical properties, such as the requirement of a stronger network sparsity condition. Besides tensor decomposition, the novelty of the proposed method also resides on a number of aspects, including the sparsity coefficient and the structure-preserving penalty term.
>
> Particularly, the proposed method considers a modified logit transformation, where a novel sparsity coefficient is introduced to accommodate the prevalent sparse networks. The sparse coefficient
> prevents the estimated entries of the embedding matrices $\alpha$, $\beta$, and hence those of $\Theta$, from diverging, even when the multi-layer network is very sparse.
>
> Furthermore, the penalty term is introduced to preserve various structures in the multi-layer network. The original penalty term is designed to preserve the community structure in the embedding vectors, which can be replaced by other regularization terms for different structures in the multi-layer network. For example, when both vertices and network layers present community structures, we may consider
> $$
> J(\alpha, \beta) = \min_\{Z, C\} ||\alpha - ZC||_F^2 + \min_\{Y, W\} ||\beta-YW||_F^2,
> $$
> where $Z$ and $Y$ are the membership matrices of vertices and network layers, respectively. When the network layers may only change at certain time points, we can consider a group fused Lasso penalty,
> $$
> J(\beta) = \sum_\{m=2\}^M ||\beta_m-\beta_\{m-1\}||_2.
> $$
> For semi-supervised node classification, we can use a different fused penalty
> $$
> J(\alpha) = \sum_\{i<j, (i, j)\in S\} ||\alpha_i-\alpha_j||_F^2,
> $$
> where $S$ contains the pairs of vertices that are known to belong to the same class. For link prediction, we can use a simple ridge penalty,
> $$
> J(\alpha,\beta) = ||\alpha||^2_F+||\beta||^2_F.
> $$
> Response to Question 1: Note that we study a multi-layer network whose layers are order free and independent. Incorporating random walks between different layers will introduce layer dependence and it is more suitable for dynamic networks, where there is a natural time order among network layers.
>
> Response to Question 2: Note that the computational complexity of the projection step is much smaller than that of the gradient step, and thus the computational complexity of the proposed method is dominated by the gradient calculation. More specifically, in calculating the gradient of $(\alpha, \beta)$, one needs to sequentially compute $\mathcal{I} \times_1\alpha \times_2 \alpha \times_3 \beta$, $\mathcal{T} \times_2 \alpha^T \times_3 \beta^T$, $\mathcal{T} \times_1 \alpha^T \times_2 \alpha^T$ and $\mathcal{T}\times_3 \beta^T$, whose computational complexity are all of order $O(n^2MR)$.  Therefore, the computational complexity of the proposed method with $T$ iterations is of order $O(n^2MRT)$, which is quite efficient as it scales quadratically with the number of vertices and linearly with the number of layers.
>
> References
>
> [1] HILLAR, Christopher J.; LIM, Lek-Heng. Most tensor problems are NP-hard. Journal of the ACM (JACM), 2013, 60.6: 1-39.
>
> [2] LYU, Zhongyuan; XIA, Dong; ZHANG, Yuan. Latent space model for higher-order networks and generalized tensor decomposition. arXiv preprint arXiv:2106.16042, 2021.

---

> > ### Author Response · Authors · 2022-08-08
> > **Further comments**
> >
> > Thank you again for your insightful comments, and we had try our best efforts to address them. Further discussions or elaborations are welcome.

---

### Official Review · Reviewer_juke · 2022-07-11

**Rating:** 7
**Confidence:** 4
**Soundness:** 3 good
**Presentation:** 3 good
**Contribution:** 3 good

**Summary:**

The paper deals with the problem of learning node  embeddings of multilayer graphs with applications mainly in community detection and link prediction. A new model is introduced, namely TLSM, that is based on flexible tensor decomposition framework. Specifically, the methodology is based on a tensor latent space model that satisfies a set of interesting properties: it allows nodes to get different embeddings even if they belong to the same community; it satisfies some key identifiability properties; and finally, it is capable of handling sparse networks though a modified logit transformation. The different claims made in the paper are supported either with theoretical arguments or empirically.

**Questions:**

The different questions that I would like to ask the authors are provided in the list of weaknesses of the paper.

**Strengths And Weaknesses:**

### Strengths:

- The paper addresses an important problem in graph machine learning, with many practical applications.
- I found particularly interesting the fact that the paper comes with a consistency analysis regarding the proposed methodology.
- The paper is also well-written, and most of the arguments made are clearly presented. I really enjoyed reading it.

### Weaknesses:

- Missing related work. Despite the fact that the multi-layer community detection literature is not as rich as in the case of single-layer graphs, still there are plenty of methodologies that follow different ideas. The paper mentions a few of them, but the literature could further be expanded. It would be interesting to also consider some of these models in the empirical analysis. I will just mention the article by Mercado et al. entitled “The Power Mean Laplacian for Multilayer Graph Clustering” (AISTATS ’18), and possibly some of the references within this article.
- I enjoyed reading the part related to the consistency analysis. However, it is not clear to me how strong are these assumptions from a practical viewpoint. How will the model empirically behave if some of these assumptions will be violated?

My main concerns about the paper are related to the empirical analysis.
- Why is the embedding dimension set to $K$?
- The scale of the datasets used is quite small. Is there any particular reason for this choice?
- There is no discussion about the time complexity of the model or even the empirical running time. I would suggest the authors to discuss this point.
- Selection of baseline models. As I also mentioned above, I found that the selected models do not cover different methodological ideas on clustering multilayer graphs.
- Lastly, despite mentioning that the paper learning embedding of multilayer graphs, most of the discussion and analysis concerns the task of community detection. There is one experiment on link prediction, but I believe this is quite limited. Did the authors think of having a more generic experimental framework that will more extensively cover the tasks of link prediction and, possibly, node classification?


### Typos:

- Line 58, *y*et
- Line  73: Greek letters
- Line 123: come*s*

---

> ### Author Response · Authors · 2022-08-02
> **Responds to Reviewer juke**
>
> Thank you very much for your  insightful comments and positive feedback. In this reply, we have tried our best efforts to address all your comments. A point-by-point reply is presented below.
>
> 1. Literature review. Thank you for bringing the related references to our attention. We will conduct a more comprehensive literature review on multi-layer network, such as multi-layer network with heterogeneous community structure in different layers,  multi-layer network with nodal covariate information, and change point detection in dynamic networks.
>
> 2. The assumptions in the theoretical analysis are mild. Assumption A is essentially not a technical assumption, but it employs a generic notation $\gamma_n$ to quantify the community separation, which will affect the convergence rate of the Hamming error. Assumption B gives a guideline on determining the tuning parameter $\lambda_n$. Assumption C is placed on the community sizes, and it is satisfied for balanced communities. Moreover, it also allows unbalanced communities if $\gamma_n$ or $s_n$ is not too small. In a way, Assumption C tries to relax the balance community assumption that is frequently imposed in literature.
>
> 3. Why is the embedding dimension set to $K$? In our numerical experiments, we set the embedding dimension to be $K$ for fair comparison with the baseline approaches, since they are primarily built upon the multi-layer stochastic block model whose spectral embedding of a vertex is $K$-dimensional. In fact, we can treat the embedding dimension as a tuning parameter and employ network cross-validation or BIC to determine its optimal value, but at the cost of increased computational burden.
>
> 4.  Scales of the real-world datasets. We agree that the scales of the real-world datasets are not large, which is mainly due to the fact that most real-world networks contain extremely sparse sub-networks. For example, the original WAT network is a $214\times 214 \times 364$ multi-layer network, but it contains many minor countries (vertices with extremely small degree) and unpopular trading products (layers with extremely sparse connectivity). These vertices and layers are removed in the pre-processing step, resulting in a $130 \times 130 \times 32$ multi-layer network for subsequent analysis. Note that these real-world applications are only used to support the advantage of the proposed method, in addition to its methodological novelty and theoretical properties. Following your comments, we will conduct different analysis tasks on a larger multi-layer network to better demonstrate the superiority of the proposed method.
>
> 5. Time complexity and empirical running time. Note that the time complexity of the projection step is much smaller than that of the gradient step, and thus the computational complexity is dominated by the gradient calculation. More specifically, in calculating the gradient of $(\alpha, \beta)$, one needs to sequentially compute $\mathcal{I} \times_1 \alpha \times_2 \alpha \times_3 \beta$, $\mathcal{T} \times_2 \alpha^T \times_3 \beta^T$, $\mathcal{T} \times_1 \alpha^T \times_2 \alpha^T$ and $\mathcal{T}\times_3 \beta^T$, whose computational complexity are all of order $O(n^2MR)$.  Therefore, the computational complexity of the proposed method with $T$ iterations is of order $O(n^2MRT)$, which is quite efficient as it scales quadratically with the number of vertices and linearly with the number of layers. In the empirical studies, the proposed method produces parameter estimation reasonably fast. For example, it takes less than 4 minutes for a $800\times 800 \times 20$ multi-layer network on a desktop computer with 3.2GHz CPU and 32G memory.
>
> 6. Baseline competitor selection. We agree that the baseline methods are rather limited, but they are chosen to illustrate the following facts. First, we would like to show the advantages of tensor decomposition approaches (e.g., TLSM, HOSVD-Tucker) over layer aggregating approaches (e.g., MASE, SEPCK). Second, we want to show the strength of TLSM over HOSVD-Tucker since TLSM does not require the column-orthogonal condition of the network embedding. Third, we also want to show the robustness of TLSM over LSE since the least square loss can be sensitive to outliers.
>
> 7. A more generic experimental framework. Thanks a lot for your comments, which indeed motivates us to think of a more general regularized embedding framework to preserve various structures in the multi-layer network. Specifically, for link prediction, we can use a simple ridge penalty,
> $$
> J(\alpha, \beta) = ||\alpha||^2_F + ||\beta||^2_F.
> $$
> When the network layers may only change at certain time points, we can consider a group fused Lasso penalty,
> $$
> J(\beta) = \sum_\{m=2\}^M ||\beta_m - \beta_\{m-1\}||_2.
> $$
> For semi-supervised node classification, we can use a different fused penalty
> $$
> J(\alpha) = \sum_\{i<j, (i, j)\in S\} ||\alpha_i-\alpha_j||_F^2,
> $$
> where $S$ contains the pairs of vertices that are known to belong to the same class.

---

> > ### Author Response · Authors · 2022-08-08
> > **Further comments**
> >
> > We really appreciate for your valuable comments as well as positive feedback, which indeed motivates us to improve the quality of the paper. We are also open for further discussion and elaboration.

---

### Official Review · Reviewer_6nBv · 2022-07-11

**Rating:** 4
**Confidence:** 3
**Soundness:** 3 good
**Presentation:** 3 good
**Contribution:** 2 fair

**Summary:**

This paper proposes a generative tensor-based latent space model, dubbed TLSM, which generates node embeddings preserving the community structure of the given multi-layer networks. Instead of directly using the log-likelihood function to encode the heterogeneous structure of the multi-layer networks, the authors also introduce a clustering type penalty to simultaneously embed the community information between nodes. Projected gradient descent (PGD) is used to optimize the model parameters, and the asymptotic consistencies of the proposed method are also analyzed.

**Questions:**

I am a bit concerned about the proposed model's time complexity. Could the authors provide more details about the model complexity?


**Limitations:**

The authors did not address the limitations and potential negative societal impact.


**Strengths And Weaknesses:**

Strengths: Firstly, the proposed model is flexible and general, with many popular network models included. Secondly, the designed regularized likelihood framework enables the proposed model to estimate the multi-layer network and conduct community detection simultaneously. Thirdly, the authors establish a theoretical analysis of the proposed model's asymptotic consistency in terms of both multi-layer network estimation and community detection.

Weaknesses: Firstly, although the proposed framework is flexible and general, the idea of using tensor decomposition is not very novel, and the regularized likelihood design is only incremental. Secondly, I think the PGD algorithm does not guarantee convergence to the global minimum, and thus we may need to select the initial point carefully. Finally, in the real-life experiments, the proposed model does not perform much better than the benchmark approaches.

---

> ### Author Response · Authors · 2022-08-02
> **Responds to Reviewer 6nBv**
>
> Thank you very much for your constructive summary and insightful comments, which will certainly help us greatly improve the quality of the manuscript. Our replies to your comments are presented below.
>
> 1. Novelty of the proposed method
>
> The proposed method builds upon tensor decomposition to incorporate the heterogeneity among layers. This is in sharp contrast to most existing methods that aggregate network layers into a single one, which may lead to information loss or sub-optimal theoretical properties, such as stronger network sparsity requirement. Besides tensor decomposition, the novelty of the proposed method also resides on a number of aspects, including the sparsity coefficient and the structure-preserving penalty term.
>
> Particularly, the proposed method considers a modified logit transformation, where a novel sparsity coefficient is introduced to accommodate the prevalent sparse networks. The sparse coefficient
> prevents the estimated entries of the embedding matrices $\alpha$, $\beta$, and hence those of $\Theta$, from diverging, even when the multi-layer network is very sparse.
>
> Furthermore, the penalty term is introduced to preserve various structures in the multi-layer network. The original penalty term is designed to preserve the community structure in the embedding vectors, which can be replaced by other regularization terms for different structures in the multi-layer network. For example, when both vertices and network layers present community structures, we may consider
> $$
> J(\alpha, \beta) = \min_\{Z, C\}||\alpha - ZC||_F^2+\min_\{Y, W\}||\beta - YW||_F^2,
> $$
> where $Z$ and $Y$ are the membership matrices of vertices and network layers, respectively. When the network layers may only change at certain time points, we can consider a group fused Lasso penalty,
> $$
> J(\beta) = \sum_\{m=2\}^M ||\beta_m - \beta_\{m-1\}||_2.
> $$
> For semi-supervised node classification, we can use a different fused penalty
> $$
> J(\alpha) = \sum_\{i<j, (i, j)\in S\} ||\alpha_i - \alpha_j||_F^2,
> $$
> where $S$ contains the pairs of vertices that are known to belong to the same class. For link prediction, we can use a simple ridge penalty,
> $$
> J(\alpha, \beta) = ||\alpha||^2_F + ||\beta||^2_F.
> $$
>
> 2. Convergence of the PGD algorithm
>
> We agree that the PGD algorithm can not guarantee to converge to a global minimum due to the highly non-convex objective function. Fortunately, our consistency results and all theoretic analysis are valid for any estimator with sufficient small objective value as quantified in Equation (8); that is, any $(\hat{\alpha},\hat{\beta})$ satisfying $\mathcal{L}_\lambda(\hat{\alpha},\hat{\beta};\mathcal{A}) \le \mathcal{L}_\lambda(\alpha^*,\beta^*;\mathcal{A}) + \epsilon_n$. For warm initialization, we employ the high-order orthogonal iteration (HOOI) algorithm to obtain a reasonable initial value, which is a popular choice for warm initialization in high-order network data analysis.
>
> 3. Performances on real applications
>
> In the real applications, the proposed method actually yields some substantial improvement over the baseline approaches in term of community detection, while its performances is still slightly better than the baseline approaches in term of link prediction. Specifically, the relative community detection errors of the proposed method over the baseline approaches are at least $\frac{18-8}{18} = 55.56\%$ and $\frac{9-6}{9} = 33.33\%$ for the AUCS and YSCGC datasets, respectively. We also remark that the link prediction accuracy on the WAT dataset can be further improved when the current community-inducing penalty is replaced by the above-mentioned ridge penalty.
>
> 4.Computational complexity
>
>  Note that the computational complexity of the projection step is much smaller than that of the gradient step, and thus the computational complexity of the proposed method is dominated by the gradient calculation. More specifically, in calculating the gradient of $(\alpha, \beta)$, one needs to sequentially compute $\mathcal{I} \times_1 \alpha \times_2 \alpha \times_3 \beta$, $\mathcal{T} \times_2 \alpha^T \times_3 \beta^T$, $\mathcal{T} \times_1 \alpha^T \times_2 \alpha^T$ and $\mathcal{T}\times_3 \beta^T$, whose computational complexity are all of order $O(n^2MR)$.  Therefore, the computational complexity of the proposed method with $T$ iterations is of order $O(n^2MRT)$, which is quite efficient as it scales quadratically with the number of vertices and linearly with the number of layers.

---

> > ### Author Response · Authors · 2022-08-08
> > **Further comments**
> >
> > Thanks again for your valuable comments, and we have tried our best to address them. We are also open for further discussion and elaboration.

---

### Meta-Review · Area_Chair_wd5Y · 2022-08-26

**Recommendation:** Reject
**Confidence:** Certain

**Metareview:**

This paper applies tensor decomposition to study the structure-preserving embedding of multi-layer networks, for the tasks such as community detection and link prediction. While the reviewers appreciate several technical novelties in the paper, they still have a number of concerns, such as the novelty of tensor decomposition in this context, the theoretical convergence, the practicality of the consistency analysis, the empirical performance of the proposed algorithm with the real-world data. Overall, the paper looks to be promising, but a little below the high bar of NeurIPS. The authors are encouraged to revise the paper based on the reviewer comments and submit the paper to the next venue.

**Award:**

No

---

### Decision · Program_Chairs · 2022-09-14

Reject